# A Novel Virtual-Based Comprehensive Clinical Approach to Headache Care

**DOI:** 10.3390/jcm12165349

**Published:** 2023-08-17

**Authors:** Thomas Berk, Stephen Silberstein, Peter McAllister

**Affiliations:** 1Neura Health, New York, NY 10017, USA; 2Jefferson Headache Center, Thomas Jefferson University, Philadelphia, PA 19107, USA; 3New England Institute for Neurology and Headache, Stamford, CT 06905, USA; peter@neinh.com

**Keywords:** headache, migraine, telehealth, virtual care, coaching

## Abstract

One major innovation, a result of the coronavirus pandemic, has been the proliferation of telemedicine. Telehealth can help solve the access problems that plague headache medicine, allowing patients in areas with no headache expertise to consult and work with a headache specialist. This is a retrospective chart review of patients seen by Neura Health, a comprehensive app-based telehealth headache center. Patients are seen by a specialist and, in addition to any medical recommendations, are given care plans individualized to their condition and recommendations at the end of their clinical appointments. The primary outcome of this study is a decrease in monthly headache days after 90 days; secondary outcomes include disability as determined by MIDAS score, depression determined by PHQ-9, patients’ utilization of emergency department or urgent care resources, as well as their global impression of improvement. The deidentified outcomes of consecutive patients of Neura Health were evaluated from March 2022–March 2023. Subjects were excluded if they did not complete all forms, or if they did not receive a clinical or coaching follow-up appointment within 90 days. A total of 186 consecutive patients at Neura Health were identified during the review period. The median decrease in monthly headache days was 55.0% after a 90 day period, headache severity was decreased by 16.7%, global impression of improvement increased by 60.9%, disability decreased by 38.7%, depression decreased by 12.5% and ER/urgent care visits were decreased by 66.1%. A comprehensive, telehealth-based virtual headache-care model significantly decreased migraine frequency, severity and disability, and is able to decrease ER or urgent care visits.

## 1. Introduction

Migraine is a leading cause of disability; headache disorders, according to the World Health Organization, are some of the most common disease conditions worldwide [1]. Headache affects nearly everyone at some point in their lives and migraine, specifically, is estimated to affect over 1 billion people globally [2]. The understanding and treatment of headache disorders has improved significantly over the past few decades, with disease-specific preventive and acute treatments now available.

Although our understanding of headache conditions has improved, and we are now able to treat them on a molecular level, the evaluation of headache disorders remains unchanged for the past 40 years. Most people with headache are not evaluated by a neurologist or headache specialist. They are often misunderstood or misdiagnosed by a primary care provider or emergency care provider. Those without access to specialty care often end up self-medicating with over-the-counter treatments that not only do not prevent the attacks from happening, but often worsen the underlying condition due to medication adaptation and overuse.

Access to specialists is mainly determined by a person’s geographic location [3]. It is much more difficult, if not near impossible, for people from rural locations, or even larger cities that do not have a headache specialist, to be seen by someone with expertise in headache conditions. Prior studies have shown significant improvement in the outcomes of patients followed by neurologists or headache specialists versus primary care physicians.

The recent COVID-19 pandemic resulted in the proliferation of virtual medical care. The use of telehealth was estimated to have grown by 15 times from 2019 to 2021. Prior to 2019, telehealth was used primarily as a means of access by patients in remote locations [4]. During the height of the pandemic, many physicians and patients used telehealth for continuity of care.

Some medical subspecialties are more appropriate for virtual care. Psychiatry has flourished in this virtual world, with many psychiatrists not planning to return to in-person visits [5,6]. Within neurology, headache medicine, and the treatment of migraine specifically is well suited for virtual care [7]. The typical age of patients with migraine and other headache conditions is the second to fourth decade, a population that is more technically savvy. This population generally consists of people that are otherwise healthy and generally have normal neurological exam findings. The majority of people with headache conditions do not require interventional therapies such as in-office injections. Telehealth can also solve the access problems that plague headache medicine and other neurological subspecialties, allowing patients in areas without a headache specialist to now have this option.

Telehealth allows a more comprehensive approach to the treatment of headache disorders. In a traditional setting, when a patient sees their doctor intermittently, the focus of the appointment is the diagnosis and medical treatment of their headache disorder. Despite the best intentions of the physician, and due to constraints beyond their control, the appointments often feel rushed, and it is more difficult to comprehensively treat patients, to fully educate them on their condition, including the appropriate non-medical options that are available for them. In a virtual setting, patient appointments are on time more often and patients can more easily be sent additional education material; in addition to their medications, patients are given an app-based, comprehensive curriculum focused on their needs. These can include neck and shoulder exercises, stress management if appropriate, relaxation strategies, and vestibular exercises. Telehealth can also improve access, as rural areas were some of the earliest adopters of telehealth appointments, well before the pandemic [4].

Despite these advantages, telehealth is not an option for all patients. Photophobia is the most common migraine-associated sensitivity, and often screen use can be a triggering activity for people with migraine. People with secondary headache disorders necessarily need further interventions including urgent imaging and a hands-on, in-person evaluation. These can often be accounted for by giving patients instructions on how to make their screens less triggering (dimming the lights, using “nightshift” mode) and by implementing strict guidelines to quickly recognize secondary headache conditions and refer them appropriately for in-person interventions.

Neura Health was founded in 2020 as the first ever comprehensive headache center that is virtual and based on telehealth. Providers are UCNS-certified or fellowship-trained MDs, or physician assistants that have extensive training and work experience at headache centers. Physicians are licensed in most states of the United States; they can offer “educational appointments” to patients located outside of those states and Neura Health providers can work together with the patient’s local physicians to provide them their prescriptions and evidenced-based recommendations. Neura’s providers follow strict protocols that are based on the American Academy of Neurology and American Headache Society’s best practice guidelines.

Few outcome studies exist for telehealth with respect to headache. This retrospective chart review is intended to help determine the effectiveness of a comprehensive approach to headache care, provided via telehealth.

## 2. Materials and Methods

### 2.1. Virtual Comprehensive Headache Center

Neura Health is a telehealth-based model of neurological care. Similar to an in-person evaluation, headache patients can make an appointment to see a neurologist or headache specialist that is either licensed in their state or that can work with the patient’s local doctors. During their virtual patient encounter, they are given a standardized virtual neurological exam, testing eye movement, facial symmetry, tongue and palate movement, drift, fine finger movement, coordination, upper and lower extremity strength, balance and gait. Medication, laboratory testing and imaging recommendations are also standardized as per the American Headache Society and American Academy of Neurology guidelines and best practices.

All patients are given care plans individualized to their unique needs at the end of their clinical appointments. These include information regarding starting or adjusting medications, and information about any prescribed medications and the underlying headache condition with which they are being diagnosed. Patients are provided with an individualized 90-day educational curriculum focusing on symptom tracking/calendaring via a proprietary headache calendar, biofeedback, physical therapy exercises specific for their specific headache condition (such as migraine, tension type or cervicogenic headache) or diet-based recommendations.

In addition, each patient is assigned a certified health coach that they can meet with on a twice-monthly basis. All Neura Health coaches are NBC-HWC trained and certified, and undergo a headache-specific training process to learn evidence-based, non-medical recommendations for headache conditions. Patients meet with their coaches for 20–30 min via video conference twice monthly, for supportive care, accountability regarding their care plans, and for goal setting based on the patients’ specific needs for the next 2 weeks. Care teams are alerted if the patient reports worsening when tracking their headaches, and work with their patients to determine what potential non-medical options may be helpful.

### 2.2. Clinical Trial

#### Subject Identification

The deidentified outcomes of consecutive patients of Neura Health were evaluated from March 2022–March 2023. Subjects with all headache disorders were included in this evaluation. An intake questionnaire is given to all patients prior to their first appointment, and patients also receive a weekly check-in questionnaire and a 90-day questionnaire. Subjects also included if they had an undifferentiated headache disorder awaiting a diagnostic test or treatment to determine their headache diagnosis.

Subjects were excluded if they did not complete all forms, or if they did not receive a clinical or coaching follow-up appointment within 90 days. An interim evaluation was performed to determine if any subject’s data were 2 standard deviations outside the mean; these subjects were excluded as well.

During their initial questionnaire, patients are asked to self-identify their current baseline frequency of headache days, severity based on a visual analogue scale, as well as the number of emergency department or urgent care visits over the past 90 days. Patients were also asked their global impression of improvement (“How close do you feel to finding relief?”). They are also given a MIDAS and PHQ-9 questionnaire to determine their baseline level of migraine associated disability and depression. Patients are asked weekly if they feel that their symptoms are improving or worsening, how severe and frequent they were on average that past week, and if they went to an emergency room or urgent care center. At 90 days, patients are asked again for their frequency of monthly headache days, average severity, global impression of improvement, MIDAS and PHQ-9.

Patients of all headache conditions and diagnoses were included, including undifferentiated headache conditions that required a confirmatory test or referral to an outside provider. These referrals were expeditiously made to local in-person providers, including ophthalmologists, endocrinologists, orthopedic or neurological spine surgeons or, if urgent, to a local emergency department. Imaging was ordered following the American Headache Society 2019 Neuroimaging Guidelines [8].

### 2.3. Outcomes Evaluation

The primary outcome for this study was mean decrease in monthly headache days after 90 days. Secondary outcomes were mean decrease in headache-related disability as determined by MIDAS score, mean decrease in headache severity, decrease in depression as determined by PHQ-9 score and decrease in emergency room or urgent care visits also after 90 days. Demographic information was evaluated including age, gender and ethnicity.

### 2.4. Statistical Evaluation

A two-tailed *t*-test was performed for all variables as noted above. The significance of alpha was determined at *p* < 0.01 with 95% confidence intervals (CI). Data analysis was performed using Big Query v3.14.0. Three predetermined interval statistical reviews were performed to recognize significant outliers, defined as outcomes more than 2 standard deviations outside of the mean.

### 2.5. Ethics Statement

This study was determined to have a D4-IRB exemption as per the WCG IRB Affairs Department. This exemption is under 45 CFR § 46.104(d)(4), “because the research involves the use of identifiable private information/biospecimens; and information, which may include information about biospecimens, is recorded by the investigator in such a manner that the identity of the human subjects cannot readily by ascertained directly or through identifiers linked to the subjects, the investigator does not contact the subjects, and the investigator will not re-identify subjects.”

## 3. Results

### 3.1. Study Design and Participants

A total of 186 consecutive Neura Health patients were identified during the review period. The data of 117 subjects were included. Exclusions were primarily due to incomplete forms at intake or after 90 days, or lack of a follow-up appointment within 90 days. Patients with secondary headaches were included after confirmation from an in-person specialist when appropriate.

### 3.2. Statistical Review

Three interim reviews were performed by a third-party data analyst. These were performed at predetermined intervals—at 50%, 75% and 90% of data collection. The purpose of the interim reviews was to exclude very significant data errors, defined as outliers 2 standard deviations outside of the mean. A total of four subjects were excluded after these analyses (Table 1).

### 3.3. Demographics

An overview of the demographics of this population can be found in Table 1. This study population had 101 females (86.3%, 12.8% male and 0.9% nonbinary individuals), and a median age of 42.0 years (14–93). An amount of 78.6% identified as Caucasian, 2.6% African American, 7.7% Asian, 0/9% Native American and 9.4% other.

### 3.4. Baseline Headache Data

The mean frequency of this study population was significantly higher than expected (18 monthly headache days). The mean attack severity was 6.5 out of 10, mean MIDAS was 62.0 and PHQ-9 was 8.0. There was an average of 0.5 ER or urgent care visits in the past 90 days, and the global impression of improvement at baseline (“Relief”) was 3.0 out of 10.

### 3.5. Primary and Secondary Outcomes

Mean decrease in monthly headache days was 29.0%, and median decrease was 55.0% after 90 days (*p* = 0, 95%CI), from 18 to 9 days per month (Table 2). Mean severity decreased 15.3%, median severity decreased by 16.7% (*p* = 0.01, 95% CI) and global impression of improvement increased an average of 60.9%; the median improvement was 100% (*p* = 0, 95% CI).

### 3.6. Other Secondary Outcomes

Mean PHQ-9 score decreased by 8.7%, the median decrease was 12.5% (*p* = 0.01, 95% CI), disability as measured by MIDAS decreased by an average of 12.8%, and the median’s decrease was 38.7% (*p* = 0.01, 95% CI). ER and urgent care visits were decreased by 66.1% (*p* = 0, 95% CI).

## 4. Discussion

Telehealth is widely used in some neurological subspecialties such as stroke to provide specialty expertise in remote settings. Over the course of the COVID pandemic, telehealth was relied upon out of necessity in many clinical settings where it was not previously used. The American Academy of Neurology has provided a position statement strongly in favor of utilizing and expanding virtual care.

Prior outcomes regarding telehealth have been rare, but positive (See Table 3). During the COVID-19 pandemic, telehealth was in its infancy, and could be optimized further with future innovations. We believe that the comprehensive nature of our care model is part of the future of headache care.

We present the first outcomes data specifically reviewing headache telehealth. This trial included patients with any headache disorder or diagnosis (migraine, tension type headache, as well as SUNCT and intracranial hypotension), all primary and secondary headache conditions. “Red flags” of headache were screened for during the virtual appointment, and any concerning finding was referred for appropriate in-person intervention or evaluation. The decision to recommend imaging was based on the American Headache Society Imaging Guidelines, in order to not over utilize imaging modalities.

This study population had a very high baseline monthly headache frequency and severity, similar to a tertiary headache center. Despite the severity of this population, there was a significant improvement in all outcomes measured. We feel that, in particular, a high severity population can benefit more from a virtual platform due to the comprehensive nature of the treatment, and the ability to avoid triggering commutes to in-person doctor’s appointments which are often triggering with bright lights, loud noises and smells.

This comprehensive approach to headache disorders has revealed statistically significant and beneficial outcomes with a decreases in monthly headache days, migraine-related disability, depression, and ER/urgent care visitation. Statistically significant, but less robust, was the decrease in overall headache intensity. We believe that this is most likely due to our inclusion of a number of refractory chronic headache disorders such as chronic migraine and intracranial hypotension. The focus when treating these chronic disorders is primarily on decreasing monthly headache days, more than improvement in the severity of each attack.

We believe that there are many significant advantages to a virtual approach to headache disorders. Many patients started seeing their medical professionals virtually over this period of time, and they appreciated avoiding the time and cost of commuting to the doctor’s office. Recent studies have highlighted the fact that most patients prefer many aspects of telehealth visits over in-person visits.

In addition to convenience, we have been able to develop a more comprehensive approach to headache care via telehealth. Patients receive a comprehensive educational curriculum, specific care plans related to their own unique issues, and individualized coaching. Although it is possible to develop similar protocols in a traditional office setting, telehealth has allowed this to be performed in a very efficient, comprehensive, and convenient process.

There are limiting factors to telehealth for headache conditions. In-person procedures and infusions must be referred to a local physician or center. Licensing across state lines can be expensive or time consuming, and state and federal regulations regarding virtual medical care are frequently changing. Providers may be concerned that although reimbursements of virtual care are currently the same as in-person appointments, this may change in the future.

Not all patients are appropriate for telehealth as well, as noted above. Patients with severe photophobia may not be able to tolerate screens at all, even with adjustments to the brightness or blue-light filtering. Patients with concern for secondary headache conditions will still need to be evaluated in person, and with appropriate referrals and imaging. Depending on the secondary etiology, they may not be appropriate to be followed long term virtually and may need an in-person specialist.

Technology is innovating much of what we do on a new daily basis. As we look to the future of medical care, we must consider ways to excel and give our patients the best likelihood for better outcomes. Leveraging telehealth’s opportunities for comprehensive and individualized care is one important step in this direction.

## Figures and Tables

**Table 1 jcm-12-05349-t001:** Demographics.

**Sex (%)**	**Female (%)**	**Male (%)**	**Non-Binary (%)**			
	101 (86.3)	15 (12.8)	1 (0.9)			
**Race**	**Caucasian (%)**	**Black/African American (%)**	**Asian (%)**	**Native American (%)**	**Other (%)**	
	92 (78.6)	3 (2.6)	9 (7.7)	1 (0.9)	11 (9.4)	
**Age**	**0–20 (%)**	**21–30 (%)**	**31–40 (%)**	**41–50 (%)**	**51–60 (%)**	**65+ (%)**
	4 (3.4)	23 (19.7)	34 (29.1)	25 (21.4)	19 (16.2)	12 (10.3)

**Table 2 jcm-12-05349-t002:** Headache frequency, severity, disability, relief and depression.

**Frequency (Monthly Headache Days)**	**Median % Decrease (Range from Baseline)**	**Mean % Decrease (Range from Baseline)**	***p* =**
	55.0 (20–9)	29.0 (17.6–12.5)	0.00
**Severity**	**Median % decrease (range from baseline)**	**Mean % decrease (range from baseline)**	***p* =**
	16.7 (6–5)	15.3 (6.1–5.2)	0.01
**MIDAS**	**Median % decrease (range from baseline)**	**Mean % decrease (range from baseline)**	** *p =* **
	38.7 (62–38)	12.8 (89.5–78.1)	0.00
**Global Impression of improvement**	**Median % change (range from baseline)**	**Mean % decrease (range from baseline)**	** *p =* **
	100 (3–6)	60.9 (3.4–5.5)	0.00
**PHQ-9**	**Median % decrease (range from baseline)**	**Mean % decrease (range from baseline)**	** *p =* **
	12.5 (8–7)	8.7 (8.4–7.6)	0.01

**Table 3 jcm-12-05349-t003:** Publications on telehealth.

Publication, Author	Year	Title	Outcome
Chiang, Halker Singh et al. [7]	2021	Patient experience of telemedicine for headache care during the COVID-19 pandemic: An American Migraine Foundation survey study	Telemedicine facilitated headache care for many patients during the COVID-19 pandemic, resulting in high patient satisfaction rates, and a desire to continue to use telemedicine for future headache care among those who completed the online survey.
Grinberg, Fenton et al. [9]	2022	Telehealth perceptions and utilization for the delivery of headache care before and during the COVID-19 pandemic: A mixed-methods study	Patients and providers were amenable to utilizing telehealth, yet also experienced technological barriers
Noutsios, Boisvert-Plante et al. [10]	2021	Telemedicine Applications for the Evaluation of Patients with Non-Acute Headache: A Narrative Review	High satisfaction rates have been reported for virtual headache management which were shown to be equal to in-person consults.
Minen, Szperka et al. [11]	2021	Telehealth as a new care delivery model: The headache provider experience	Respondents were comfortable treating patients with migraine via telehealth. They note positive attributes for patients and how access may be improved.

## Data Availability

The data presented in this study are available on request from the corresponding author. The data are not publicly available due to HIPPA and the potential of patient identification.

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
