# Peer review of "A Novel Virtual-Based Comprehensive Clinical Approach to Headache Care"

_jcm, 2023, doi:10.3390/jcm12165349_

Round 1
Reviewer 1 Report
This paper is about a novel virtual clinical approach to headache. There are some major concerns I have regarding publication.
- The statistics section requires more information, as how do you normalize the data? Which program do you use? Is there a reason you didn't use correlation to analyze your data?
- For me it is not clear what your pipeline is. It is not clear what your goal is, nor is it clear how you are interfering. Explain why you chose MIDAS and PHQ-9 and what their purposes are.
- Why did you not use the MADRAS scale, which is more accurate and accepted?
- In your discussion you mentioned that there was a significant improvement in all outcomes. It is unclear, compared to what? How did telehealth interfere in this result?
Reviewer 2 Report
Few areas of concern
1. Article states: Within neurology, headache medicine is well-suited for virtual care, as most patients are otherwise healthy, have normal neurological exam findings, are technologically savvy and do not require interventional therapies such as in-office injections.
Comment:
A. Headache Medicine is a subspecialty concerned with the diagnosis and treatment of head and face pain. Headache medicine specialists also provide for painful disorders of the head and neck treating disorders such as Trigeminal and Occipital Neuralgia, Cervicogenic headache, and secondary causes of headache such as a infection, vascular disease and exposure to environment sources.
Ref: https://www.mcw.edu/departments/neurology/patient-care/clinical-specialty-programs/headache-medicine
Suggest rephrase: Migraine medicine is well-suited for virtual care, as most patients are otherwise healthy, have normal neurological exam findings
B. Most headache patients are ‘Technologically savvy’
Comment:
Is there a statistic to back up this statment?
Also, counter-intuitively:
(i) Rural and less technology enabled areas which are the most benefited by expert advice.
(ii) Increased use of phone/tablet screen is a common trigger for a common condition such as migraine and also avoided by those with photophobia.
(iii) The following reference by the authors themselves, state that patients and providers experienced technological barriers.
Grinberg, Fenton et al [8] 2022 Telehealth perceptions and utilization for the delivery of headache care before and during the COVID19 pandemic: A mixed-methods study Patients and providers were amenable to utilizing telehealth, yet also experienced technological barriers.
C. Do not require interventional therapies such as in-office injections
Comment:
This depends on the setting. As such, headache clinics provide subspecialist care, are referred by other patients with complex headaches, requiring multidisciplinary management including in-office injections.
2. Often the appointments are rushed, and patients are unable to be educated on their condition, or what appropriate non-medical options are available for them. In a virtual setting, patients can be given additional education material, in addition to their medications.
A. Often the appointments are rushed, and patients are unable to be educated
Comment:
The corollary implies that those practicing in-patient visits are either unethical by spending lesser time, or being forced to be at rush, and not able to spend the time that is required to be actually spent.
B. ‘patients are unable to be educated’
Comment:
Why? Is it because they don’t want to be educated, or the doctor does not want to educate them?
C. In a virtual setting, patients can be given additional education material, in addition to their medications.
Comment:
Why is this not possible during in-patient visits in a headache center with a team (like the virtual team) at work?
3. Patients are provided with an individualized 90-day educational curriculum focusing on symptom tracking/calendaring via a proprietary headache calendar, biofeedback, physical therapy exercises specific for their specific headache condition (such as migraine, tension type-or cervicogenic).
Comment:
Physical therapy for cervicogenic headache should only be safely prescribed:
A. If cervical spine is stable. This requires clinical examination and testing which cannot be done virtually.
B. Specific Exercise programs should be supervised by qualified persons to avoid injury.
4. Subjects with all headache disorders were included in this evaluation
The data of 117 subjects was included. Exclusions were primarily due to incomplete forms at intake or after 90 days, or lack of a follow up appointment within 90 days.
Comment:
Secondary headaches such as cervicogenic, intracranial hypotension and SUNCT have been included in the author’s cohort. These require diagnostic imaging, diagnostic blocks and, in case of SUNCT, cross consultation with ophthalmology and/or ENT evaluation, all of which need physical visits.
Further comments:
1. Although the aim is to look at the improvements in patients conditions, the authors have not mentioned how many required physical visits for a neurological, head and neck exam. This has relevance because the article is trying to mention the benefits of virtual consultations in headache medicine and not just migraines.
2. If secondary headaches were excluded,
A. without physical examination, it could mean underestimation of secondary headaches and,
B. If bypassed physical examination, it could mean over-utilization of investigations.
C. If they have been included when red flags were identified, the article fails to mention how how many patients required physical visits and how many required investigations to confirm the diagnosis and/or to exclude sinister conditions, which is important before promoting the pros of virtual headache consultations.
3. Overall, this is not a balanced article as attempts to mention several limitations of both the study and of virtual consultations have not been made before concluding that ‘We believe that there are many significant advantages to a virtual approach to headache disorders’.
Recommendations:
1. There should be a primary physical contact for the patients to be examined and, to be safe.
2. Diagnosed cased of primary headaches may be followed up through telemedicine.
3. Virtual consultation may have been the only option during the pandemic, but in post pandemic time, at least one visit should be a physical visit.
Few typos such as:
1. In a traditional setting, a patient sees their doctor intermittently the focus of the appointment is the diagnosis and medical treatment of their headache disorder. This sentence when corrected, possibly reads: In a traditional setting, when a patient sees their doctor intermittently, the focus of the appointment is the diagnosis and medical treatment of their headache disorder.
2. Full stops and commas are missing in one or two places.
Round 2
Reviewer 1 Report
I agree with your arguments.